# Paradoxes of Emotional Life: Second-Order Emotions

Antonio de Castro Caeiro

Research Grup on Ancient Philosophy, CultureLab, Nova Institute of Philosophy (IFILNOVA), NOVA FCSH, Universidade Nova de Lisboa (UNL), 1099-032 Lisbon, Portugal; cajc@fcsh.unl.pt

**Abstract:** Heidegger tries to explain our emotional life applying three schemes: causal explanation, mental internalisation of emotions and metaphorical expression. None of the three schemes explains emotion though. Either because the causal nexus does not always occur or because objects and people in the external world are carriers of emotional agents or because language is already on a metaphorical level. Moreover, how is it possible that there are presently emotions constituting our life without our being aware of their existence? From the analysis of boredom in its three varieties ("bored by X", "get oneself bored", and "it is boring") we will get to the depth where emotions lie, trying to rouse them and to keep them awake. Although it surfaces with the force and energy of the present, every emotion has its past and future constitution. How can we understand the future of a present emotion along with its past?

**Keywords:** emotional life; deep psychology; anonymous intentionality; subconscious life; depth emotional life; boredom; angst; anxiety; time; metaphor; causality; motivation; Wallace Stevens; Max Scheler; Martin Heidegger

## 1. Some Puzzling Questions

(1) An emotion can be located deep inside the mind. (2) Emotions are inner reactions caused by events situated in the outside world. (3) Emotional reality is indirectly expressed by metaphors [1]. (4) We really need to feel what is up and what is going on to get emotional [1] [2]. One can easily validate these theses unpreparedly. However, on closer inspection, the inside-outside criterion to locate emotions is not very accurate. In the opening line of Wallace Stevens' poem "The house was quiet and the world was calm" [3], it is the house that is quiet and the world which is calm [2] [4]. Quietness and calmness really exist in the outside world, not inside my head or, more enigmatically, in neurons or synapses. When one experiences a "summer night" one does not need to know anything about neurology or neurophysiology [3] [5].

Quiet and calm can transfigure an entire evening, but is something inside my mind going outside of my mind to metamorphosize the external world? The thesis presented in (2) deepens (1). Can we establish a causal relation between an object (a thing, a person, a landscape, an event) in the outside world and an emotion inside ourselves? [4] [6] The same object does not always cause the same emotions. Emotions have different shades and several degrees of intensity. Different objects can cause the same emotion in us. At different times, the same object can cause different emotions in the same person or the same emotion in different persons. The same object can even cause a deep emotion in someone and go completely unnoticed by someone else. The thesis in (3) implies that we understand things and persons, allowing us to shuttle between sense and reference, between meaning and facts. There is more about this further down in this paper.

The literal objective reality of facts can be expressed in figurative language. We can export our subjectivity to the outside world. We can also import objectivity into our inner world. We already live inside an atmosphere of meaning that admits these apparently disparate and irreconcilable languages. Talking about "quiet house", "calm

world", "thinking cigarettes", "cosy rooms", "inhospitable cities" is possible because objective real facts and subjective figurative meanings are different ways of expressing the same reality, i.e., life. Is it possible as (4) states, that there may be emotions that are not felt now while being present. How come? This seems contradictory and paradoxical. How can it be that an emotion is and is not simultaneously present? However, is it not true that there are emotions and sensations that may be building up now, as we have an experience, without being felt? Looking back at any given past moment of our lives, we can feel the emotions we went through without having felt them when they occurred. How is it possible that remembering our high school time we feel all those emotions, feelings, sensations and dispositions that we must have felt then, but did not identify as we carried on with life at that time. Even now, at this very moment, emotions are being formed in a dimension in which we do not feel them. Emotions are not as clear-cut now as they will be sometime next year, perhaps while visiting the same place. Present emotions will arise in the near or distant future as the emotions we are "getting through" now without feeling them.

Life happens in an emotional environment. Emotion is neither interior nor exterior, it is neither objective nor subjective, or it is both subjective and objective. An emotion can be cause and effect. It can be present and not be felt. It can be absent and be felt. It can be associated with an object or a person. It can happen on its own, without any apparent object. Emotions are both private and public, personal, and collective [5] [7,8]. Even though they can happen momentarily, the impression they make can last forever. Emotions are the ways we feel about life and anything that happens to us. There is, therefore, a spectrum of infinite possibilities of emotions. There are levels of depth. In vol. 20/30, *The Fundamental Concepts of Metaphysics* [9], Heidegger isolates one fundamental attunement–boredom at three completely different levels of depth. We will follow his introduction to the phenomenon, where Heidegger discusses these paradoxical theses. We will try to reach a more comprehensive understanding of the possibility of second-order emotions (4.) [6] [10,11].

Heidegger is a name for a phenomenological operator. I put under scrutiny a set of phenomena that Heidegger analyses. I do not intend just to churn out a textual exegesis of Heidegger. The aim is to isolate the specific form of phenomenological openness to the emotional dimension of life. By understanding the situation in which we find ourselves every time already as an emotional situation, it is possible to access phenomena that lie in the same constellation as those that Heidegger presents. They are not, however, the same. If, on the one hand, the idea is to open up the emotional dimension in order to access deep emotional phenomena, on the other, the purpose is to show how the structuring depth of the emotional level makes this same opening possible. The emotional level that we access in an incipient way when we try to isolate first-order boredom may be the same one that is being anonymously and sub-consciously retroactively projected to make those inaugural steps possible. Or not. We may not recognise the deep level that is retroactively projecting itself as resulting in our interest in analysing emotions. For example, our philosophical or psychological curiosity to perceive phenomena such as boredom, fear, anguish, melancholy, and so on is the subconscious being of the emotions. The very being of the emotions is the background that articulates each emotion. The whole emotional level may be a horizon that we know alongside the cognitive and voluntary levels. The being of emotions may be the background that articulates each emotion. The whole emotional level may be a horizon that we know alongside the cognitive level and the volitional level, but we do not understand the unity of meaning of the emotional neither its relation to cognition and volition, nor the possible subconscious or unconscious relations, the short-term reach of an eruptive emotion and the long-term, existential, reach of emotions, the kind of emotions that are definitive and have ontological characteristics that are life defining: the phenomenon of interest, the possibility of love, the passion for truth, the religious scruple, our exposure to the sublime, and so forth and so on.

Heidegger is a name for such a phenomenological operator. The examples are not just instantiations of an emotional form, but correspond to experiences in the first person singular. Now, the reader is also a first-person singular and will be able to find the phenomenal basis for understanding what is at stake each time. The balance between the presentation of an autobiographical testimony, an interpretation of an example given by an author who may also be an autobiographical testimony or just an example, is difficult. Moreover, while thinking of the reader, I cannot let him be confronted with auto or heterobiographical examples without him thinking of his own personal experiences. On the level of the emotional example, the private character can only be "destroyed", "deconstructed" and broken down if it has occurred to a person. To identify an experience, we must already have been in a similar situation. Sometimes it may happen that we are in denial of some emotional experience. We may think that we have been in the same situation, or in similar (but not quite the same) situations. Even the protagonist of existence, which is each person in their existence, has very diverse ways of being in the same situation: at a birthday party or on Christmas Eve, with the same people, there are experiences so disparate that they seem to be the experiences of completely different people and yet we are the same "person", it is me that has been there all along. Yet it seems to me that I am almost a different person at each birthday party or on each Christmas Eve.

On the other hand, the formalisation of an abstract thesis—"formally the emotional experience is different from person to person"—does not allow us to understand in a concrete way what this difference from person to person consists of, or what this difference is in various moments of the same person's life. Even if one recognises the same kind of emotional experience, the same form, circumstances make the difference. Even this thesis that the same emotion is different at various levels of depth is experienced differently by several people or by the same person over a lifetime; maybe an abstract thesis cannot not lead us emotionally to the emotional experience of this diversity. Abstract theses about emotions can be understandable if, and only if, the reader has the key to their de-formalisation. Now, the hypothesis that there are different types of otherness requires the passage from the abstract to the concrete, from the impersonal to the personal, from the public to the private. Nothing can facilitate understanding more than "invoking" situations that we went through or opening ourselves up to the possibility of having the experience that is being proposed to us as the only one that demonstrates with the force of evidence what is at stake. No one will know what boredom is unless they have experienced some degree of boredom.

Emotion is a tone, or cadence, that vibrates deep in our lives. It has a real function. No emotion is blind. It offers an understanding of something about something, about someone, about ourselves. Even without our necessarily doing anything about it. We simply feel the power of emotions as feelings articulated in language and interpretation. Every emotion has a script, if we can call it that. We interact with emotions because we do not unequivocally understand what they mean. We just know that they mean something or tend to understand emotions as phenomena that are trying to say something to us. They mean something. The atmosphere we live in, our most intrinsic environment is emotional.

The normal understanding of emotions interprets each emotion (1) as the mental effect of an 'exciting' agent in the outside world; (2) as a mental phenomenon within the *psykhê*; (3) as a metaphorical expression. Furthermore, (4) to become "emotional", emotions must rise to the surface of consciousness. We suggest that this is the other way around. In fact, we will first begin with the reversal of (4). We suspect that emotions may be present, conditioning our lives at any given moment, without "declaring" their presence to us. On the other hand, some emotions do make their presence known but are superficial, they vanish as soon as they show up. Emotions constitute the ultimate frontier of our conscious life. They admit depth. If so, can we go deep enough to get a glimpse of this emotional dimension where meaning, sense, and understanding of what is going on may be available? If so, do theses 1–3 collapse?

Heidegger's analysis of boredom in its deep fundamental dimension will allow us to understand the different layers and degrees of emotional experience. We know too well, as a matter of fact, that we do experience boredom in some situations. This empirical level is enough for a first characterisation of this emotion on a superficial level. From then on, we will try to understand how we can get to a deepening of boredom, specifically, and of other emotions or dispositions on dimensions that at first glance, and for the most part, are removed from our normal everyday experience.

So, what is decisive for us is to understand how boredom can emerge as a fundamental deep emotion. Can boredom, like any other fundamental emotion, be happening without our having a perception of it. Let us therefore try to understand (1) how the cause-effect relationship fails; (2) how boredom is so external that it characterises a being in the external world (a person, a situation, a moment in our lives); and (3) whether the metaphor is already the primordial expression of our relationship to life (towards the world, others, ourselves)? Does this mean that even facts are already expressed through metaphors and analogies? It may happen that meaning antecedes facts, and sense overshadows reference. Therefore, metaphors express the a priori emotional grounding of all our life experiences.

Our take is, therefore, that there are levels of emotional experience, ranging from concrete situations full of emotion to dimensions we can only reach after sufficient time has elapsed for us to understand emotions that were present then at that past time [7] [2]. Does this mean that we are going through emotional layers in our present moment of life that we can only experience as unambiguously emotional in the future? How can we have the foresight of the emotional forecasting of our future?

## 2. Boredom as a Fundamental Emotion

How, and why, does Heidegger suggest that boredom is the emotion that attunes our life? Out of all our emotional experiences assessed as positive and negative, why does he stress deep boredom? Why not, say, love? We do identify things, people, situations, phases, and periods in our life as boring, but how is it that boredom lies as a deep fundamental emotional disposition at the bottom of all beings? So, we understand that some beings are boring. Sometimes we are boring, but can we say that we react to an anonymous presence of boredom all the time in our lives? Can a fundamental emotional disposition be constitutive without showing up all the time? Can we be bored without "feeling" boredom? Would it be possible to fill up our lives with all sorts of activities we enjoy performing all day long and be bored? This amounts to feeling emotions (enjoyment, pleasure, fullness) completely different from boredom, but let us not rush in dismissing Heidegger's thesis as not making sense yet. We do have the perception of moments in time that are interpreted as setbacks and delays. We get stuck in moments. Time seems to freeze. Sometimes, we do experience feelings of emptiness, situations without any meaning. Are these moments in which boredom manifests itself enough for us to go deeper into the depths of that dimension where emotions lie without declaring their presence to us? Can we go down to such depths? How? Could boredom emotionally attune our personal and collective lives? Is Heidegger's analysis circumscribed to a time, almost one hundred years ago, and should it be dismissed on historical sociological or even psychological grounds? Do not our lives in the 21st century seem so much better and full? Or can boredom still be alive getting us both at a personal and collective level? Can we vibrate with joy, loving what we do, and at the same time be bored to death? The diagnosis of our current situation stems from a sensation and, as such, it is a short-lived phenomenon.

> "Das Ganze ist eine Sensation, und das heißt immer eine uneingestandene und doch wieder scheinbare Beruhigung, wenn auch nur literarischer Art und von charakteristischer Kurzlebigkeit. (Everything-in-its-entirety [das Ganze] is a sensation [Sensation], and that always means an unacknowledged and yet apparent serenity, even if it is only of a literary nature and characteristically short-lived.)" [8]. [12]

We get into a situation in which impressions caused by something or someone are felt, leaving us in a certain state of mind. Can something or someone be the cause of the sensation we feel? What if the sensation is caused by us and then spreads to everything, over a period? For example, on a Sunday afternoon, even for a fleeting moment, everything seems to be boring. This sensation disappears as quickly as it arises. We do not know where it came from or where it went. We do not know why boredom showed up, but we know that an entire city can be engulfed by this feeling of ennui that pervades everything. Is boredom asleep just to wake up on a Sunday afternoon?

Emotions are at the basis of the philosophy of culture. Its aim is a diagnosis of culture (Kulturdiagnostik) [9] [12]. Now, every diagnosis presupposes a prognosis, "it can constitute and become a prognosis (zur Prognose aus- und umbildet)" [10] [12]. The philosophy of culture seems to have good intentions. Knowledge of our personal and collective past, nationally and worldwide, seems to allow us to get to know our present better so that we can understand where we are going, but words such as "diagnosis" and "prognosis" seem to suggest a civilisational disease. Are we in danger as a species? When one talks about survival, the planet's resources are not the case in point. The survival at stake is none other than that of existence itself based on the emotional dimension where all our aspirations and expectations take place.

Heidegger's perspective is different than that of the cognitivists. His philosophy is much closer to Kierkegaard and Nietzsche than to that of the philosophers of emotions. For Heidegger's commentators, he belongs to that 19th-century tradition which operated the inversion of the hierarchy attributed to the psychological acts. Therefore, while tradition has privileged representational acts (logos) over volitional (ethos) and emotional (*pathê*) acts—a thesis popularised by the Stoics and upon which even Kant bases his philosophy, Heidegger reversed that order. However, he does not say that first is the emotional, then the volitional, and only thirdly the logical or representational. He says what Scheler and Husserl had already intended. He bestows a noetic dignity upon synthetic acts of the mind (synthetischen Gemütsakte). With phenomenology, love ceases to be blind. Heidegger seeks to situate the revelation of being, the truth of being, on a plane prior to that of representation itself. The question of the meaning of being is posed by the discovery of dispositions grounded on a plane of openness and closure that is that of the situation in which each of us finds ourselves. Reflection and self-perception can completely block my access to myself. On the other hand, I can absolutely wish to hide from myself and not succeed, because my most irrational fears find me, the most sordid dreams find me, and the most abject thoughts find me.

Therefore, the emotional is not mental content, nor a *noêma* or a "representational" content, but the reality as such. I do not love the mental content of my mother. I love my mother. The relation between myself and my mother is already emotional. Why duplicate the objects? What Heidegger seems to do is to get rid of those presuppositions that do not help but rather impair any comprehension of emotions of what is already happening.

The majority of Heidegger's analyses from the point of view of the hermeneutic tradition, even when they stress the affective or dispositional turn, consider emotion as a regional aspect of an ontology. It is not clear, however, how one releases a personal emotion—an affective crisis with emotional impact, for example—in order to interpret it in a philosophical dimension. Even when one goes through a heart-rending experience of romantic (*erôs*) or religious (*agapê*) love, there is an enormous difficulty in understanding how others can go through the same thing or how these feelings could both have been at the basis of all the stories of romantic love and lead the Christ to the cross. From the point of view of the non-continental tradition, Heidegger is interpreted, sometimes sympathetically, as advocating fundamental aspects of pragmatic philosophy and clearly adding to the emotional plane, which can be blind and mechanical, or merely a "cognitive" (instinctive) aspect. Now, Heidegger criticises the primacy of the cognitive as access to the world. In his view, in-Sein is affective and not cognitive. This aspect of openness and access to the totality of being is obscured by both the hermeneutic and the analytic or even pragmatic traditions

in their approach to the emotions. Heidegger's philosophy is a "Stimmung". It is from the interpretation of *thauma* and *thaumazein*, or philosophy as *erôs* and as *pathos*, simple and absolute, that Heidegger must be approached. In this same way, the erotic experience in the platonic sense of the term is an experience of the maximisation of our personal being (in our relationship with things, with others and with ourselves), because we have access to an absolute, maximum, superlative version of ourselves. It is the superlative, emotionally open, version of ourselves that is looking at us now, here in the present, from the future. The absolute exponentiation of myself puts enormous pressure on the version of each of us now. This tension results from an ulterior version, possible but effective. Each of us projects ourselves towards this superlative possibility of our own self, open ourselves to the other, seek another to love, seek God in religion, seek the sublime in art. Thus, for Heidegger, being and the truth of its meaning, the problems of philosophy, open up in the affective tone of an emotion. Being opens up in moments of truth and revelation emotionally. The emotional plane is the agent of *alêtheia*. The experience of being is always emotional and affective. It happens to us on the plane of everyday life because such visitation is possible. What we need are eyes to see it. There is thus always an exposure and a vulnerability in everyday life to this ecstatic, existential dimension of emotion. Existence in Heidegger means: to be [continuously] going outward [from within], to manifest oneself. However, if the temporal organisation can naively be: past, present, future, in Heidegger it is inverse: future as possibility (possible or simply impossible) (Entwurf), present (Verfallenheit), past (Gewesenheit). It is the future that goes out of itself and approaches us in the present, in a "movement" of ever smaller and smaller inflows and ever larger and larger outflows. The flow of existential time and the structurally temporal disposition is an ever-smaller influx of possibilities coming from the future, and an ever-larger flow of lost possibilities in the past. Centring Heidegger's analysis on these aspects leaves Gadamer's hermeneutic interpretations in *Wahrheit und Methode*—the matrix text of Hermeneutics—far short of what Heidegger intends. Damasio's analysis of the emotions, for example, is blind to the total and ontological dimension of the problem of being and the truth of being that Heidegger seeks to pose [11] [2,5,12–17].

This is the reason why Heidegger speaks of the awakening of a (not "the") fundamental disposition. What if boredom is what emotionally, and therefore existentially, constitutes our lives? "It is precisely this the reason why we are striving to awaken a fundamental emotion (gerade wenn und weil wir die Weckung einer Grundstimmung erstreben)" [12] [12]. However, awakening is different from observing. "There is a theoretical difference between observing for cognitive purposes our spiritual situation and awakening a fundamental emotion (Es besteht ein theoretischer Unterschied zwischen der Darstellung der geistigen Lage und der Weckung einer Grundstimmung.)" [13] [12]. This means that our situation may be symptomless. The emotional symptoms can be ambiguous. Do we need for the main character of one's life, oneself, to get "sick" with boredom? For if we do not feel boredom, we need to stimulate its presence. We should at least try to remember any boring past situation and dive into it to understand what has happened.

What is valid for boredom is valid for all emotions. All emotions have a layout linking their unmistakable surface eruption and their deep existence. In that sense, "we would not be allowed to ask where are we? We should rather ask how it is with us? (dürfen wir nicht fragen: wo stehen wir?, sondern müssen fragen: wie steht es mit uns?)" [14] [12]. How can we delve into the moments in time when we were bored, when time had stopped and everything was empty? At that time, we cannot ask: "I wonder if what happens to us after all is that there is a deep boredom in the depths of our existence that, like a silent fog, pushes and pulls us wherever it goes? (Ist es am Ende so mit uns, daß eine tiefe Langeweile in den Abgründen des Daseins wie ein schweigender Nebel hin- und herzieht?)" [15] [12].

What do people, things, situations have in common when they all become boring? What do books, shows, evenings, holidays have in common to be or become agents of boredom? Is it because "we ourselves get bored because we become bored with ourselves? But must the human being himself become bored with himself? Why is this so? (Etwa weil

wir selbst uns, uns selbst, langweilig geworden sind? Der Mensch selbst sollte sich selbst langweilig geworden sein? Warum das?)" [16] [12].

We need to cast off all theories. It is through direct observation that we know what happens when we are bored: time stops, and one feels emptiness. Does the depth of a disposition directly correspond to the power with which it appears and the scope it has? We tend to think that the stronger the emotions are the more real they are, but could an emotion, despite not being felt, exist and exert pressure upon our life, even without our being aware of it? How do we know whether this emotional dimension really and effectively exists if it is not manifest? Perhaps we can understand its anonymous presence through our way of being, acting, in our attitudes and behaviour. We avoid certain situations because we think they will bore us, but how is future boredom captured? It is undeniable that there are people and situations that cause us boredom. What is manifested in this direct contact with boredom is that time slows down and everything seems empty and superficial. We can now find stimuli that move us or interest us.

> "We had just asked if it happens to us after all that a profound boredom moves back and forth in the abysses of existence like a silent fog. (Wir haben nur gefragt: Ist es am Ende so mit uns, daß eine tiefe Langeweile in den Abgründen des Daseins wie ein schweigender Nebel hin- und herzieht?)" [17] [12]. "What does it mean: is boredom problematic for us? First, formally, it says as much as this: we do not know if it conditions us emotionally now. (Was heißt: die Langeweile ist für uns fraglich? Zunächst sagt das formal soviel: Wir wissen nicht, ob sie uns durchstimmt oder nicht.)" [18]. [12]

(1) This emotional feeling of boredom has no reality. Yet, it may be having an effect. How can a phenomenon like a disease exist causing effects without symptoms? Its manifestation seems to be fully armoured to us. (2) Boredom as emotion is metaphorically described as a haze. Unlike a thick fog, it lifts and disperses. It is concentrated in certain places and very light and tenuous in others. It sways as the wind blows. "In the end, we do not want to know anything about that emotion, but we are constantly trying to avoid it. (Wir wollen am Ende nicht von ihr wissen, sondern suchen ihr ständig zu entgehen.)" [19] [12].

Like all fundamental emotions, profound boredom can exist in a deep, unfathomable dimension. However, it can erupt from time to time in episodic moments. We may even not realise that it is boredom. We do not even have a name for this outbreak. Thus, it may be that we know what boredom is because it has already been present in our lives with its devastating power. Based on those past experiences, we do not want to know anything about it, let alone awaken this emotion in order to interpret its deep wisdom. Boredom is not a pleasant feeling. Maybe all deep dispositions have this uncanniness about them, we do not want them to break free from wherever they are kept. However, we understand that we are continually reacting to them, anticipating their presence. We may try to always be busy, have things to do, close ourselves off from the manifestation of those emotions that are coming from that deep dimension of existence.

> "How to escape boredom in which, as we say, time becomes long? We are simply striving so much, consciously or unconsciously, to pass the time, that we welcome the most important and essential occupations, just because they fill our time. Who wants to deny that? But then, is it still necessary to ascertain that this boredom is there? (Wie entgehen wir der Langeweile, in der uns, wie wir selbst sagen, die Zeit lang wird? Einfach so, daß wir jederzeit, ob bewußt oder unbewußt, bemüht sind, uns die Zeit zu vertreiben, daß wir wichtigste und wesentlichste Beschäftigungen begrüßen, schon allein, damit sie uns die Zeit ausfüllen. Wer will das leugnen? Bedarf es dann aber erst noch der Feststellung, daß diese Langeweile da ist?)" [20]. [12]

We need to kill time, we want "to kill time", we want time to pass. We tend to be running away from boring situations, trying to escape them when they are there, but we know that boredom can come at any time. How come? What grounds this knowledge

about this emotion? We know very well that boredom can always come again. Like all deep and profound emotions, they are asleep for the most part of our lives. We all know what depression, melancholy and sadness are when they are present, but we also know that, somehow, they can and will show up again in our lives. They vanished but they did not go away forever. We know they can come back. We know this because in the past they had disappeared but then reappeared again. Every episode of deep emotional experience brings with it interpretive intelligibility. Emotions are intelligible. They allow us to understand what is going on with us when they manifest. Even if they do not release a full-fledged knowledge of ourselves, we know all deep emotions have something to say about us. It is from the future that deep emotions come to us. When in a deep phase of depression, we may get out of bed without feeling the presence of depression. We have our breakfast; take a shower. We might wonder why depression has not yet arrived. It feels like we are anaesthetised. However, when dealing with anxiety and depression, we know it is only a matter of time, and then it comes. Anxiety or boredom attacks us. What happens between the moment we get out of bed and the moment we feel the presence of anxiety? Is this absence of feeling a depressive emotion the same as when we do not feel boredom? Is it different [21] [18]?

"But what does it mean: we expel and drive boredom away? We always make boredom fall asleep. (Was heißt das aber: wir vertreiben und verscheuchen die Langeweile? Wir bringen sie ständig zum Einschlafen.)" [22] [18]. Killing time, occupying time, filling up time has a clear and unambiguous meaning: to anaesthetise us against the uncomfortable presence of boredom. We may even come across tasks, occupations, that have become mechanical or automatic as a pragmatic reaction to the presence of boredom. When we feel boredom kicking in, we try to get busy, we keep ourselves occupied, as a sort of self-conceived occupational therapy, but does free time expose itself to boredom? Is boredom already ever-present, waiting for the right moment to show up? With what intention?

"We 'know'—what a remarkable thing to know it—that boredom can always return at any moment. (Wir "wissen"—in einem merkwürdigen Wissen –, daß sie doch jederzeit wiederkommen kann)" [23] [18]. This knowledge of how it is with us (in relation to an emotion) opens up the possibility of being affected by profound dispositions. The way emotions appear and disappear, rise and fall, hover above us and harass us is intrinsic to this phenomenon. We avoid boredom and situations that we think will bore us. However, where does boredom as such come from? The same question can be asked regarding anxiety, anguish, fear, melancholy, and sadness. No emotion of this deep kind disappears forever.

> "So, boredom is already there then. We try to kick it out. We try to put it to sleep. We do not want to know anything about it. This does not mean that we do not want to be aware of it, it means, rather: we do not want to keep it awake—this boredom that is after all already awake and with its eyes wide open—even if absolutely keeping its distance—looking into our Da-sein from the outside and penetrating and tunin g us with its look. (Also, ist sie schon da. Wir verscheuchen sie. Wir bringen sie zum Einschlafen. Wir wollen von ihr nichts wissen. Das heißt ja gar nicht: wir wollen kein Bewußtsein von ihr haben, sondern es heißt: wir wollen sie nicht wach sein lassen—sie, die am Ende doch schon wach ist und mit offenem Auge—wenn auch ganz aus der Ferne—in unser Da-sein hereinblickt und mit diesem Blick uns schon durchdringt und durchstimmt.)" [24]. [18]

### 3. Emotional Stalking

Heidegger avoids talking about consciousness or unconsciousness as far as our attunement to emotions is concerned. Being aware of an emotion is totally different from waking it up or not letting emotions slip out of our minds, or even putting them to sleep. We say we do not want to think about emotions whose content causes us anxiety or suffering. Yet, that is all we think about. It imposes itself. The opposite can also happen. Sometimes we would like to "feel again" what we had felt in a certain situation. However, those past feelings and emotions do not show up again to be "lived" by us. To trigger a past emotion

is to tune into it, but we can wake up emotions that were not even felt in past situations. There is, however, a faint vibration of this past manifestation in the present. We cling to this emotional thread so that past emotions and feelings can fully manifest as they were. By so doing we try not to let sentimental references to the past "die" entirely or at least not let them fall asleep. There are emotions, affects, feelings that exist as living characters. Some are like people dear to us whom we love. There are also sinister characters who inhabit the attic or basement of our lives. They are clandestine but we know perfectly well that they cohabit with us. Usually, we do not come across them. Yet, we feel that these emotional characters follow us at every step. They are constantly stalking our existences. They know everything about us, what we do, what we think, who we are, how we are. Feeling the gaze of the emotional disposition from the outside, at a distance, affects us. This is how the look of emotional depth feels like. We feel vulnerable to that gaze of emotions. We are transfigured by the look of emotional damage. Now, it is precisely this transfiguration that we do not want to let fall asleep. We do not want to escape from it. On the contrary, we want to be running towards that emotional feeling that lets us know something about ourselves. All we need to do is strive to keep that emotional depth awake [25] [19]. Maybe we can then learn about ourselves from it. Emotions do say "something" about us. They are agents of truth.

> "But if emotions already are awake, then they do not need to be woken. Not really. Awakening this fundamental disposition does not mean waking it up first, but letting it stay awake and preventing it from falling asleep. We can easily infer from this that the task has not become any easier. Perhaps this task has become much more difficult; perhaps because we know that it is always easier to wake someone up with a shock than to prevent them from falling asleep. (Aber wenn sie schon wach ist, dann braucht sie doch auch nicht geweckt zu werden. In der Tat nicht. Das Wecken dieser Grundstimmung heißt nicht, sie erst wachmachen, sondern wachsein lassen, vor dem Einschlafen behüten. Wir entnehmen hieraus leicht: Die Aufgabe ist nicht leichter geworden. Vielleicht ist diese Aufgabe wesentlich schwieriger, ähnlich wie wir jederzeit erfahren, daß es leichter ist, jemanden durch einen Schock aufzuwecken, als ihn vor dem Einschlafen zu behüten. Doch ob sie schwer oder leicht ist, das ist hier unwesentlich.)" [26]. [19]

So, the task is finding the fundamental emotion that, by vibrating, makes us understand what goes on with us at the bottom of our lives. This emotional depth is at first shielded. It is seen as always existing vibrantly. It "exists" from afar but close enough to watch our life at every moment. Fundamental emotions and feelings have already surfaced. They came and went, after being alive for a lapse of time. The difficulty, then, is to recapture in some way what feelings let us feel. What happened when some emotions were present, affecting all our life with their presence: the world, others, ourselves. How can we reconstruct the story of an emotion and our dealing with it? How can we resuscitate what it made us feel, the impression it made? How are we to measure its emotional impact? What is the state of mind it left us with? If there still lives in us an emotional hint of what went on, we do not let it fall asleep, even if all that is left of that emotion is only a shred of life. However, it is very hard to know how to keep awake an emotion that wants to go to sleep. Or is it me that somehow wants some emotions to go to sleep?

"The task, then, is not to let boredom go to sleep (Die Langeweile nicht einschlafen zu lassen)" [27] [12]. From the outside, this methodological task seems to counteract our tendency in life to thwart and resist motionless moments of time, setbacks, and delays. We always want life to keep on going. Yet we know all too well about those moments when we feel bored and are stuck in them.

When a circumstance of boredom forms, we try to "kill time", to chase away and dissipate the oppressive presence of boredom. We try not to allow it to be there. When it comes to pass, we do not want it to stay awake, we try to put boredom to sleep ("die Zeit vertreibt und die Langeweile gerade nicht aufkommen läßt, das heißt, wenn sie kommt,

sie verscheucht, sie zum Einschlafen bringt?)" [28] [12]. Now, however, "we have to keep boredom awake (Wir sollen sie wachsein lassen.)" [29] [12]. We still do not know how, because this move is against the natural tendency of life, which is to try to escape boredom, by finding a way to occupy time, by finding occupations so as not to let life "stand still". So, we chase boredom away. Now we want to do the exact opposite, but how to proceed? Whenever we feel bored, should we just let it endure and try not to move so that boredom grows? When boredom arises, will we have enough presence of mind not to react, to try not to think about something else? However, to think about what is happening to us, does it not disturb the very presence of boredom? Can we, by remembering situations of boredom, bring back the state of mind in which we lived them and thus release the disposition felt in the past so that it can attune us to it?

> "Boredom has a varied multiplicity of figures (Gestalten) that we know all too well in their most diverse disguises and masks (Verschleierungen). When it emerges (auftaucht), it affects us in the blink of an eye, for a moment, or else tortures and afflicts us for long periods of time. As soon as it appears, it is there, we try to repress it, we strive to expel it from our lives ("Die Langeweile—wer kennt sie nicht, wie sie in den verschiedensten Gestalten und Verschleierungen auftaucht, uns oft nur für Augenblicke befällt, oft auch längere Zeit quält und bedrückt. Wer weiß nicht, daß wir, sobald sie kommt, uns auch schon daran gemacht haben, sie wegzudrücken, und bemüht sind, sie zu vertreiben)" [30]. [12]

Are there other forms of boredom so profound that they are not identified as boredom, at least most of the time and at first sight? Is it possible to go through vibrant situations as opposed to the monotonous cadence of boredom, and still identify them as boring, for example, when having fun or working enthusiastically? Having fun or working hard can be ways of spending time, but they can denounce that we are bored, or otherwise we would not try to escape. Can fun and work be just the surface of boredom?

> "Or is this boredom that we know and of which we now speak in an indeterminate way a mere shadow of true, genuine, authentic boredom? In fact, we ask and continue to ask again and again: is it really happening to us that deep down there is a deep boredom in the abysmal depths of our existence pulling us this way and that to determine our own existence? ("Oder ist diese Langeweile, die wir da so kennen und von der wir jetzt so unbestimmt sprechen, nur ein Schatten der wirklichen? Wir fragten ja und fragen immer wieder: Ist es am Ende so weit mit uns, daß eine tiefe Langeweile in den Abgründen des Daseins wie ein schweigender Nebel hin- und herzieht?)" [31]. [12]

There is a relationship between boredom and time. The duration of boredom episodes can be short or long. However, the time of emotions does not make them exceptional. Everything has a beginning, middle and end in time. What makes the relationship between boredom and time so special? Boredom lengthens time. When it happens, there is a metamorphosis of time in us. The time of life, which passes continuously without us always being aware of it, emerges to the surface. Even any superficial manifestation of boredom allows us to understand the essential relationship we have with the time of existence. This relationship with deep time cannot be undone.

> "Boredom (Langeweile) indicates almost palpably, and especially in our German word, a relationship with time, a way in which we situate ourselves in relation to time, a feeling of time (Langeweile ... zeigt fast handgreiflich, und besonders in unserem deutschen Wort, ein Verhältnis zur Zeit, eine Art, wie wir zur Zeit stehen, ein Zeitgefühl. Also führt uns die Langeweile und die Frage nach ihr zum Zeitproblem.)" [32] [12]. " ... or is it the other way around and boredom only leads us to time, to an understanding of how time vibrates in the depths of existence and of how we can, therefore, 'act' and 'maneuver' alone in our usual superficiality? (Oder ist es umgekehrt, führt uns die Langeweile erst zur Zeit,

> zum Verstehen essen, wie die Zeit im Grunde des Da-seins schwingt und wir
> deshalb in unserer gewohnten Oberflächlichkeit allein "handeln" und "lavieren"
> können?)" [33]. [12]

## 4. The Four Paradoxes

It is now time to return again to where we left off. Starting now from the interpretation of boredom, can we understand how the three schemes for explaining emotions do not apply? (1) The cause-and-effect relationship (das Ursache-Wirkung-Verhältnis); (2) the subjective interiority of emotions (das Innerseelische); (3) the metaphor (Übertragung) as an expression of emotions; and (4) the reductive manifestation of emotion blocking the experience the deep emotional level. Schemes 1–3 do not even apply to the superficial level where emotions show up, but let us apply the three schemes to boredom. The starting point should be to identify the active element of boredom in a boring situation (im Ausgang vom Langweiligen). When we feel we get bored by something, we can identify the active element in something or someone that bored us in each situation. On the other hand, we are bored because we are liable to getting bored. At last, the synchronicity of the active and passive elements produces the whole situation of boredom. As we have seen, once in a situation of boredom time slows down and it comes to a halt. One feels it is never time. Time stands still. We are kept on hold, waiting (das Hinhaltende). The second elemental characteristic is the feeling of emptiness. We are deprived of sense and meaning mainly because in that situation there is nothing telling us anything in order to fulfil our being there (das Leerlassende). In this sense:

> "das Verstehen der Stimmung verlangt von uns am Ende einen Wandel der Grundauffassung des Menschen. Die rechtverstandene Stimmung gibt uns erst die Möglichkeit, das Da-sein des Menschen als solches zu fassen. (Understanding of emotions requires of us, after all, a transformation of the fundamental understanding of being human. Emotions correctly understood finally give us the possibility to grasp the Da-sein as such of the human)" [34]. [12]

Our relationship to the active element of boredom in an object is a relationship with that boring object and not a relationship to our representation of the emotion "boredom" [35]. [12] Heidegger stresses again and again that when we feel bored, it is due to the relationship with an object really existing in the outside world and not with the emotion felt in the stream of consciousness, inside our mind.

> "Langweiliges kennen wir so, weil es in und durch seine Langweiligkeit in uns Langeweile verursacht. (We know the "boring", the active element in a thing, because the quality boredom as such is what in its essence and through itself makes us bored)" [36] [12]. What can "boredom" be? "Etwas Langweiliges—ein Ding, ein Buch, ein Schauspiel, ein Festakt, aber auch ein Mensch, eine Gesellschaft, aber auch eine Umgebung oder eine Gegend—solch Langweiliges, das ist nicht die Langeweile selbst. (Everything can be boring—a thing, a book, a show, a party, but also a person, a company, but also the surroundings or a region)" [37]. [12]

There are three figures of boredom that Heidegger analyses. The concrete experience of boredom can be totally passive. The formulation is in the passive voice and accentuates the boring agent, that which leaves in us the sensation of boredom: "Von Langweiligem werden wir gelangweilt, so daß wir uns dabei langweilen." (We are so bored by the boring that we are bored with it)" [38] [12]. There is something or someone, a situation, circumstance, that "causes" an impression on us and leaves us in a boring state. This is being bored "by" (das Gelangweiltwerden durch ein solches Langweiliges). The second form of boredom is reflexive: getting bored by a situation (das Sichlangweilen bei einem solchen). The third is: boredom as such (die Langeweile selbst)" [39] [12]. This structural analysis identifies several possible figures of boredom, which go far beyond the subject–object relationship, given

their complexity. We can understand the different manifestations of boredom using the passive voice (to be bored and be bored by), the reflexive voice (to be bored with) and the active voice: (to bore, boredom). Boredom is a *nomen agentis* expressing the intransitive verb to bore. By identifying this figure, Heidegger aims at presenting several complex forms of experiencing boredom for which the subject–object relationship and the philosophical classification ideal and real, empirical and rational, are short-sighted categories. In all three forms, the starting point is always the concrete experience of boredom, though. Each figure stresses one aspect of boredom. The same object does not always have to be boring, nor do we always get bored with that same object.

## 5. Emotions Outside and Things Inside?

"Die Langeweile ist nicht einfach ein seelisches Erlebnis im Inneren, sondern etwas von ihr, das Langweilende, was das Sichlangweilen entspringen läßt, kommt uns gerade aus den Dingen selbst entgegen. Die Langeweile ist viel eher draußen, sitzt im Langweiligen, und von draußen schleicht sie sich in uns ein. (Boredom is not simply a psychic mental experience inside our minds, but there is something in boredom, the boring itself, which triggers boredom, and which comes to us precisely from the things themselves, in the outside world. The boring element in things comes from the outside and insinuates itself into us" [40]. [12]

The boring thing is the thing (Ding) out there: the people themselves, the events, shows, landscapes, regions, dust from the house, temperature, everything that is out there. There is not an object and a boring element in it. It is that thing that is boring. This discovery shifts the centre of gravity in the analysis from the mind to the things themselves. Indeed, boredom is neither in the chemical elementary structure of a spatial-temporally determined and specified thing in its corporeal matter, nor is it found in neurons or synapses as such. That is why the experience of something or someone as boring is the original starting point from which we begin our analytical thinking. It takes place before any theory that one may have about phenomena of this nature, for we have always known what it is like to deal with boring things and persons in boring situations. What is decisive here is the concrete experience of emotions at play. So, we need to know how to deactivate all explanatory theories about emotions and feelings based in science. All discoveries come later than the phenomena. We do not need to wait for science to understand the powerful effect of an emotion. What do we find boring in a person, in a show or in a book? They drag us on and dry us up. The boring exists in the book in its relationship with us, in seeing it as an object and in reading it. The boring drags on and is arid. By dragging on, it slows us down. When we find something as boring, we say that it drags on. Life is put on hold. At the same time, it is arid and dry. It contaminates us. Makes us empty.

"Langweilig—wir meinen damit: schleppend, öd; es regt nicht an und regt nicht auf, es gibt nichts her, hat uns nichts zu sagen, geht uns nichts an. (Boring: what do we mean by this: it drags on, is arid; neither stimulates nor provokes anything, gives nothing of itself, has nothing to say to us.)" [41]. [12]

## 6. Causal Relation

After the identification of things outside of us, it is important to understand the causal relation between things and us in order for them to be understood as boring. What do they provoke in us? How do they cause us to experience boredom [42] [12]? The boring "ist das, was uns langweilt, also Langeweile verursacht. (The boring is that which bores us, therefore that which occasions boredom.)" [43] [12]. "Was das Langweilige in seiner Langweiligkeit ist, können wir doch nur aus der Langeweile verstehen, und nicht umgekehrt. (Only from boredom are we able to understand what the boring is in its boring quality and not the other way around)" [44] [12]. Boredom is the meaning from which we are able to identify boring objects there in the real world of life outside ourselves and, conversely, emotions of boredom. The verb "occasion" (Verursachen) can be read as causing an effect, but it never

recognises a mechanical or physical causal relationship between an object in the outside world and an emotion inside our minds. It is not like when the temperature drops and the thermometer shows the mercury falling. Or when a billiard ball sets another ball in motion, diverts the course of another ball, or stops it [45] [12]. To occasion means to propitiate, to bring together the conditions for something to happen, even though it may not happen.

> "Denn was heißt es, gewisse Dinge und Menschen verursachen in uns Langeweile? Warum gerade diese Dinge und jener Mensch, diese Gegend und nicht eine andere? Ferner, warum dieses Ding jetzt und ein andermal gerade nicht, und was früher langweilte, plötzlich gar nicht mehr? Es muß doch an all dem etwas sein, was uns langweilt. Was ist es? Woher kommt es? Was uns langweilt, sagen wir, verursacht Langeweile. Was ist dieses Verursachen? (For what does it mean that certain things and people cause boredom in us? Why precisely these things and that person, this region and not another? Moreover, why this thing now and not some other time and what about what once used to be boring and is no anymore. There has to be something in all of this that bores us. What is that? Where does it come from? That which bores us, we say, causes boredom. What is this occasioning?)" [46]. [12]

To have an application of the category of causality, conditions must be met so that when A happens, B happens. Now, this is precisely what does not happen. The fact that we are bored with the contents A, B and C or X, Y and Z does not mean that every time A, B, C, X, Y, Z occur we feel bored. It may happen only once. It may be episodic, sporadic. It can also happen that what did not previously cause boredom, now becomes deadly boring. The opposite can happen. A person annoys you and then ceases to be annoying. We do not really know what is it in these objects that cause us to be bored when they provoke boredom and then disappear to no longer cause boredom. This set of problems makes it impossible to draw a causal link between the occurrences of A to Z and tedium. We can positively say that when one feels bored, a relation is established between A, B, C, or X, Y, and Z. The boredom is a mood or an atmosphere that we associate with an object, a person [14]. No question about it. The thing about this experience is the identification with its temporality: it drags on and on, time seems not to elapse because it is dry, it is empty and leaves us feeling empty. Any dull object has this relation with time and with emptiness: it is a retardation of life, it forces us to be dragged through time, and it is empty, it does not fulfil, it does not interest, etc. "Langweilig—wir meinen damit: schleppend, öd; es regt nicht an und regt nicht auf, es gibt nichts her, hat uns nichts zu sagen, geht uns nichts an." (Boring—we mean by this: it drags on, it is boring; it does not stimulate and stir, it provides nothing, it has nothing to tell us, it does not concern us.)" [47] [12].

The boring drags us along and wears us out. In another formulation: it stops time and is draining, but these are still characteristics that we sense in us. They concern us. They alter us. Are they subjective? The interpretation of the subjectivity of these characteristics does not cancel the objectivity of their existence in things and in people that are boring when they annoy us. Subjectivity is not understood as something inner, impermeable, absolutely shielded. There is an interplay between mental reality and objective reality just as there is a personal interaction that admits dispositional transformations between people. We are affected by others as others are affected by us. This a priori atmosphere priori is already open as such and is as much objective as it is subjective, or neither objective nor subjective. Any exclusive characterisation of this emotional phenomenon would clearly not take into account what happens here. Just as we do not have a representation of boredom that is projected onto content to render it boring, we also do not have an actual imported boring thing that makes us feel bored. The same thing happens when someone annoys us or is always annoying to us or we annoy somebody or are boring to this person. There is not an interplay of boredom representations between us and others. We instantly feel boredom, that we are being bored or being boring. The emotion is already in the air if you will. Boredom is a possibility and that is the reason why time is experienced unfolding

with content, but that can change. It can stop, it can be held back, it can be delayed. It can be emptied of its meaning. This interplay between person and thing, or between persons, is always just downstream of the opening to the existential horizon in which we live with emotions, affections, commotions, mental states, moods and dispositions with their cadences, vibrations, rhythms, and times.

It is important to stress the phenomenological origin of the analysis again. The progress of the analysis is slow but steady here. We are dispelling misunderstandings. The cause-effect relationship, the internalising of the mood as a mental phenomenon, and the metaphor are all operators that multiply the problems, when it is about the possibility of tuning into the mood as it is happening, i.e., trying to understand what it is telling us about the reality, our own selves, our relation to what happens to us in the emotional milieu. So, the starting point is the dispositional vibration that is felt, the affect, the emotion, the pathos, or whatever we may call it:

> "Wir sagen: aus einer Stimmung, aber nicht einer verursachten Wirkung; aus einer möglichen, uns möglicherweise befallenden Stimmung. Aus einer Stimmung her finden wir etwas so und so und sprechen es so an. Das heißt nicht: eine Wirkung und ihren Charakter auf die bewirkende Ursache übertragen. (We are saying: from an emotional cadence. Not, from an effect caused; a possible disposition that could possibly happen to us. From an emotional dispositional cadence, we find that something is like this and we talk about it. This does not mean: transfer an effect and its features to the effector cause.)" [48]. [12]

## 7. Metaphors

Heidegger's third characteristic, which he refines, is the metaphorical. When we talk about dispositions are we speaking about things or metaphors of things? It is the same thing when we use allegories and parables. Words can either denote or connote. We know that, but experiencing that ground of the connoting or metaphoric usage of a word is ineradicable. The literal meaning of a word is one thing; the figurative use of a word is something else entirely. Yet when we speak of a dull person or a dull book, what are we saying? Is it not true that we are exporting mind phenomena and contents of a dispositional nature to persons and things? Further. The impressions that things and persons cause on us and the state they leave us in is then transferred to things and persons. There is a transmission chain that allows for dispositional reception of contents, emotional clipping of them, and then, at the appropriate time, we use that content to dispositionally describe what appears.

> "Langweilig, heiter, traurig (Ereignis), lustig (Spiel)—diese stimmungsmäßigen Eigenschaften, sie sind im besonderen Sinne subjektbezogen; nicht nur das, sie stammen direkt aus dem Subjekt und seinen Zuständen. Stimmungen, die die Dinge in uns verursachen, übertragen wir hinterher auf die Dinge selbst. (The attributes "boring, serene, sad, funny" are emotional, they exist in a relation to the subjects in a particular sense. Not only this. They directly originate something in the subject and its states. The dispositional qualities that things cause in us are transferred by us to the things themselves.)" [49]. [12]

Heidegger does not criticise metaphor as an expression of emotion. He is rather trying to show how the fundamental element of the understanding of meaning is always already implied in the metaphor. Therefore, if there is an excess of meaning expressed through the use of metaphor, it is not the figurative meaning that comes after the literal, but the inverse. The figurative sense is primary in the very constitution and expression of meaning. It is not a simple matter. What Heidegger seems to be trying to say is that metaphor in the strict sense already presupposes the use of language, which is in its original sense metaphorical, allegorical, a parable. In Portuguese, the word for "word" is a translation of "parable". The literal corresponds only to a suspension of the second-order use of the metaphor which prevents the "figurative" use in the strict sense. Is scientific language, even mathematics,

capable of a literal use of language? Does not what is equal, for example, or reflexive or transitive presuppose a sense other than the literal based on a world in which no two things can be presented as equal? Meaning and its expression are a priori in relation to which there can be literal and figurative senses, denotation and connotation. One does not start from a literal sense to a figurative sense, because the literal sense itself presupposes access based on the understanding of the sense itself. That is, what can be understood as a literal understanding is still metaphorical, an angle from which reality is seen, but which is not without equivocity. Now, the relation between meaning and reference can lead, on the one hand, to the neutralisation of meaning in order to stay only with the referent. After all, a = b if "a" and "b" have the same referent. "The evening star" is "the morning star" if, and only if, they have the same referent because "afternoon" and "morning" are completely different senses. On the other hand, a triangle is the same as a trilateral, although an angle is different from a side and only the 'tri-' makes it possible to understand that they are the same object. The fact of signification is once again the a priori, just as the figurative was first in relation to the literal [50] [20].

A smiling meadow, a serene room, and a melancholic landscape allow the disassociation of objects from geography, architecture, and painting to be addressed according to the moods they may arouse in us at any given time. The cut-out smiling, serenity, and melancholy can mutatis mutandis be applicable to other objects according to the manner and mode of being of these same objects [51].

## 8. Second Order

Analysis indicates a possibility that might pass unnoticed were it not to be taken up explicitly in the following paragraphs. It may well be that an emotional mood is there constituting both the reality of something or somebody boring and the subjectivity of the condition in which we each find ourselves. Yet, it may well happen that we do not realise its presence. "[es] ist sehr wohl möglich, daß wir uns beim Lesen gar nicht gelangweilt haben, nicht das "Gefühl hatten", daß in uns Langeweile bewirkt werde. ([it is] very possible that we were not being bored while reading, did not "feel" boredom being induced into us.)" [52] [12]. I may read a boring book. Yet, to read it, I have to somehow be pulled into the reading, be concentrating and understanding the main thread. A dull book does not necessarily prevent one from reading it. The same thing happens when watching a movie, a play, when visiting a museum, going to the beach, walking around, or meeting somebody. We might not realise that each one of these contents is boring. We have the experience of time dragging on, appearing to stop, and being delayed. On the other hand, it does not fulfil us, it is not full. It is, instead, an emptiness. We have dealt all our lives with tedium, that time that drags on and on to the point that it seems to stand still and to be a delay of life. We understand how it feels to cope with content which evacuates any sort of filling: "Aus einer Stimmung, von der wir dabei wissen, daß sie jederzeit aufsteigen könnte, die wir aber niederhalten, nicht aufkommen lassen wollen. (It is starting from the dispositional cadence, which we know could arise at any moment, but that we want to repress, that we do not want to let emerge.)" [53] [12].

Yet, we unmistakably know that something was boring back then. It was boring to read that book, to watch that spectacle; entire quarters of a city, or an apartment, or the décor of a home, but also times of the day, or days, or phases of life, or epochs were boring. The thing common to all boring things is that they drag on, they stop and drag us with them, they bring us to a standstill, they retard our life. Moreover, they are dull, they are boring, they do not stimulate, they have no interest. "Schleppend besagt: es fesselt nicht; wir sind hingegeben, aber nicht hingenommen, sondern eben nur hingehalten. Öde besagt: es füllt uns nicht aus, wir sind leer gelassen. (Dragged means: it does not hold our attention; we give ourselves away, but we are not taken, we are kept on hold. Desert means: it does not fulfil us, we are left empty)" [54] [12]. That which is drawn in comes to be translated ontologically as what delays (das Hinhaltende) and what is depicted as desert: what empties (Leerlassende). How we find ourselves, and the dispositional openness to

this cadence, enables us to identify content as boring and the state we find ourselves in as boredom: "wie wir so und so angegangen wurden und uns dabei so und so befinden. (the way we were affected and how we find ourselves affected)" [55] [12].

Thus, the longer or shorter duration of an emotion cannot be the criterion for deciding on its importance. It may happen that we have no other way of accessing the deeper dimension of emotions than when they surface consciously within a period. "Perhaps that boredom that so often slips by us is more important than the one we strive to annihilate" (Vielleicht ist gerade jene Langeweile, die oft nur gleichsam an uns verbeihuscht, wesentlicher als die, mit der wir uns gerade ausdrücklich abmühen)" [56] [12]. The fact that "it leaves us in an unpleasant and uncomfortable situation (in ein Unbehagen versetzt)" [57] [12] may not mean anything. Neither duration nor emotional violence are necessarily criteria of truth. "Perhaps that boredom is more essential when it does not make us feel good or bad, but rather leaves us as if we were not even under any emotional presence (Vielleicht ist jene Langeweile wesentlicher, die uns weder gut stimmt noch mißstimmt und doch stimmt, aber so, als seien wir überhaupt nicht gestimmt.)" [58] [12]. "This superficial boredom must lead us to profound boredom, or, to put it more properly, superficial boredom must reveal itself as deep boredom, by tuning us to the depths of existence. This passing, casual, non-essential boredom must become essential. (Diese oberflächige Langeweile soll uns gar in die tiefe Langeweile bringen, bzw., angemessener gesprochen, die oberflächige soll sich als die tiefe Langeweile offenbaren, uns im Grunde des Daseins durchstimmen. Diese flüchtige, beiläufige, unwesentliche Langeweile soll wesentlich werden.)" [59] [12].

The relationship between surface boredom and time may not be immediately identified. However, when we get a brief glimpse of the phenomenon, we immediately understand this relationship. In these situations, we try to kill time, occupy ourselves with tasks. We feel that time is slow to pass. This lets us understand we get a feeling of time or, rather, time makes itself felt. Time "tells us" about itself by lengthening or shortening its pace, producing in us the sensation that it passes quickly or slowly. However, the time of boredom is different from the time when we cross episodes of boredom. The time of boredom is the time of existence itself. It never ends as long as we are alive. Surface boredom comes and goes. The time of authentic boredom comes from the depths of existence. What if the time of our lives was but an occupation of the free time of existence?

The relationship between surface and depth is very clear here. By putting the surface in relation to the background, we can understand that it is from the depths of existence that the condition for the possibility of experiencing surface boredom is constituted. The episodes of boredom depend upon deep boredom. Depth here means the a priori and transcendental condition of the being of boredom. It may happen that the ontic truth of boredom is circumscribed and isolated without allowing us to understand its profound dimension. The usual criteria must be put under this perspective. What is fleeting, constantly flying away, on the surface, is so determined in contrast to the background, which is the entire time of existence. Superficial emotions can indicate their depth. The time from which they come is the time of life that we carry along the way. This time already existed when we came to life. Perhaps, then, everything is "inside" this time, which has existed at every moment of our lives. To understand what this time has to say to us, we need not resist it as soon as it emerges (Nichtalsogleich-Widerstehen) but let the emotions and feelings that accompany this time vibrate in its authentic cadence (Ausschwingen lassen.) Ibid.

The emotional a priori allows us to find ourselves. The phenomenon indicates an openness [21] that is different from the reflective one or the one given by self-perception. The experience of feeling emotions opens oneself to others [60] [22,23] the world, oneself [61] [24]. Not only that: time is the source of all our emotions, feelings, and moods. Is time an emotion? How can we have a perception of time if not through feeling it, through a sensation of time [62] [25,26]? The causal explanation fails, because sometimes it works, sometimes it does not. There are many situations in which we do not identify any emotion at play. Looking back, though, we can have an insight into the shape of the emotion that was at

work then. This second-order emotional dimension must be active for us to act the way we do. All our attitudes and behaviours are expressions of the emotional dimension that does not show up but is active in a clandestine way [27]. We know only too late what kind of emotional dimension (of feelings, passions, affects, moods) was active and working us out at any given moment in time. Somewhere, sometime, next year or in years to come, we will experience the constellation of emotions, moods, and feelings that are constituting our lives in this present moment [28]. How can we actively get there, instead of just passively noticing this fact. What does this mean? Does the emotional dimension of our lives come from another world, from the future, is it teleological? How does it hurl itself on us? Does this dimension hoover above us to let itself be discovered? How?

## 9. Emotional Depth

There are no emotions that are exclusive to the past, emotions that are exclusive to the present, and emotions that are exclusive to the future. Emotions are not, as we have seen, just reactions to actions with corresponding responses. There is anticipation in emotions. They are pro-active, they open perspectives for the future. Without emotional perspective, the future is also cancelled. There are, without a doubt, emotions that give or seem to give more importance to the past: nostalgia for bygone times. In the present, we feel clearly cut stimuli: tension provoked by appetites of all kinds: hunger, thirst, addictive contents, but also sexual tension, irritation, and fury. In the present, we are exposed to all kinds of emotional stimuli. Even the memory of an episode from the recent past can disturb us strongly. We are exposed to all kinds of emotional provocations. Finally, hope and despair are emotions clearly grounded in the future: the excitement caused by the moment of anticipation, the promise of pleasure, and the threat that one feels coming from imminent danger.

Still, it is possible to understand the temporal shifting character of emotions not only as phenomena that take place in the past, present and future, but as events that take place with the past, the present and the future. On the other hand, the duration of an emotional episode can never be circumscribed by a chronometer. Although there are chronic depressions, there are also more or less sad and joyful phases in our lives that last for days, weeks, months and even years. Difficult mourning can last much longer, but there are also emotional phenomena that correspond to an epiphany, they allow for a turnaround in what seemed to be the set meaning of the course of a lifetime. This is how we find love or break up effectively with someone forever. What happens in an hour can affect our lives forever.

The radical origin of the deep emotional level is the future as possibility. We all have great hopes for our lives. We live with great expectations. This is the level of depth that gives rise to the apparently anonymous and subconscious nature of emotions. Trying to make them reveal themselves is the very work of the philosophy of emotions, and perhaps of philosophy as such.

A fundamental aspect in which I distance myself from Heidegger is his dogmatic use of the thesis of the three emotional levels: first-order or superficial, second-order or more profuse, and third-order or depth. To begin with, it is not evident that there are only three orders and that the combinations are not more unclear and ambiguous. Moreover, one does not perceive the exclusive focus of the analysis to be on boredom. There would be the possibility of irradiating to other dispositions, or at least of referring the surface-depth structure to other dispositions. Heidegger refers to boredom as a fundamental emotion, but not as the only one. *Being and Time* lists guilt, bad conscience, death, fear and anguish as fundamental dispositions. However, the point to underline is this: from its deepest to its most superficial level the emotional plane reveals time in its happening. Time is emotion (which seems a little more bizarre and difficult to understand). While at the first level we can apply causal relationship to perceive boredom, or any superficial sensation, as a response and reaction to what happens to us, the same is not true of a second and third-order emotional response. Moreover, while in a second-order emotion I do not perceive

the emotional structure that I later come to realise as the meaning of the situation at the moment I was experiencing it, a deep or third-order emotion sheds light on the future temporal element as preponderant. Still, there is multiple combinations of emotional levels that we experience in a single day, without realising why we go through these emotional moments. We have the perception that this is so. So, we try to understand why we feel the way we do. It is precisely when it is difficult to understand the meaning of our emotional situation, that is, when we do not understand why we find ourselves in it, that we try to find the key to that understanding. We can have a picture of this in an episode of William James' situation of unrest described by Scheler [63] [29].

On certain weekday afternoons, James was required to give a course on formal logic. Logic, especially formal logic, was not even remotely an interest of William James'. Now, one particular morning, he noticed quite early on that he was feeling especially moody, "jumpy", with an eagerness to go about doing this and that without being able to concentrate on a specific task. He was, as it were, experiencing a sort of self-prescribed occupational therapy. James paced back and forth in his office and all around the house, collecting bits of paper, sharpening pencils, sitting down and getting up, and so on. Then, he tried to figure out the reason for his state and wondered if it could be due to having read late into the night. He knew perfectly well that he was going to teach the class that day in the afternoon, but it did not even occur to him that the state he was in was the result of an appointment, of a class scheduled for a certain time, a future time relative to the early morning of that same day. Where does the psychological "influence" of emotional states we often find ourselves in come from if not from the future? Are there not countless examples of similar situations that we all experience? Scheler examines the case further. It is not just an isolated instance of something we do not want to have on the agenda that makes a bad impression on us and leaves us in a bad state. It is the whole indeterminate future that is creating pressure on us, but we cannot then make a mathematical induction: if it happens that I am in a bad emotional state because of a future event this means that all future events cause a precarious emotional state. What happens is the opposite. It is because there is a future that all time can already constitute itself now, just as a moment ago, before that, yesterday, the day before yesterday. The future as possibility, beyond any limit, is ever since emotionally and anonymously structuring my whole life. Most of the time, and primordially, I count on "my future" without any reflexion upon it. However, when experiencing despair and anxiety, or dull emptiness, the whole time of the future is cancelled and its cancellation already makes its effect felt now. We often say that one person has a future and another has no future at all, and we know how good prospects bring joy and brightness, thus colouring all our experiences, and how bad prospects cloud everything. We do not need to have any clear representation of what is going to happen, of what our perspective on reality is. However, the fact is this: we are subject to a perspective that opens up from the future; there is a retrospective that comes from our future and that makes our whole future, our whole present and its content, our whole past and its content, our whole life and its content bright or sombre. So, there is also a prospect already at work, even if we do not realise it, opening up to a future moment in time, even if we do not live to see it happening. A recalcitrant experience allows us to perceive the future of an emotion. In surviving trauma, we understand that everything is going to be different from then on. When old age strikes, when we understand an episode that makes everything irreversible, we get a glimpse of the never-again, we understand what forever means.

Or rather, I now have all the time in the world, all the time in the universe, all the time of eternity. Yet I cannot make anything of it. What does it mean not to be able to make anything of time? It means not knowing how to fill it or else how to kill it, how to make myself not feel unoccupied for all the time of my life. What deep boredom indicates when it appears is the same as the anguish of death. Boredom empties and paralyses, it brings emptiness and inanity with it. It interrupts my life abruptly and totally. It is that definitive, empty feeling that envelops all my emotions, all the moments across which my life is distributed. Boredom exerts pressure on us and manifests itself in such a way that we

know we are having a bad time. Deep boredom turns every day of our lives into a Sunday afternoon with nothing to do. Our whole life is that Sunday afternoon. We have all the time in the world and do not know how to enjoy a single moment of it. Is there no emotion that can rescue us from this tedious moment? Is the emptiness of the future the dimension that runs parallel to all the instants of our life with or without felt emotions? There is not a moment in our lives that is not founded on this emotional depth that overflows outside the boundaries of my life and reaches into others' past and future lives. Permeating everything with its powerful "no", this sense of emptiness that comes from being bored makes us ask ourselves the question of meaning. How is it with you? Have you been living a good life? Has it been worthwhile?

The sub-conscious emotional level is detected in the present moment when one feels discomfort, malaise, restlessness. However, the emotional provocation comes from the future. Now, how is it that a felt emotional phenomenon is the effect of a cause that lies in the future. Is this not an inversion of the natural understanding of causality? Is not the cause in the past? Is not the effect of a cause its consequence and its future? Emotional depth comes from the future. The emotional foundation lies in the future. The emotional situation we are in comes from the future. It is in the future that we have to look for the reason why we are like this. The answer can be found in the fact that in (just) a moment we are going to go through a situation that is already stressing us. Yet we are not having any representation of that future scene. Without a thematization or representation of what is going to happen, or of how the future is going to happen, we are already emotionally metamorphosed. What is harder to understand is that, as a future, an emotion is a mere possibility. Yet it is a possibility that can be far more effective than any reality.

We feel the pressure of the hour when we will be doing something we do not want. We know quite well what we have scheduled. It is different from the indeterminacy of the future of a Thursday at 5 p.m. for which we have not scheduled anything, or from my present discomfort at the idea of teaching 6 p.m.–9 p.m. classes in September when it is still August. We are thus always already under the pressure of a future moment. For Heidegger, the fundamental question is that somehow deep boredom transforms life. "Sunday afternoon in a big city" is an expression of emptiness, of inanity, of the total suspension of time in my life [64] [12]. When something like this happens, it is not only my inner mental psychology that is transformed, but the whole "world", the whole "universe", everybody else, everything else. Life itself shows up and reveals itself. It is a moment of revelation, a moment of being. Maybe now we can ask "what is the meaning of being?" or is it the other way around? Is it not that Being Itself asks us: what is up? What is going on? What are you up to in your life? One needs to get close to those phaenomena. Sometimes deep emotions surface in our conscious life. It is a matter of fact, but the work of philosophy is to make them show up and to allow us to live in such a dimension where being unleashes itself emotionally.

**Funding:** This work was funded by national funds through FCT—Fundação para a Ciência e a Tecnologia under the project UIDB/00183/2020.

**Institutional Review Board Statement:** Not applicable for studies involving humans or animals.

**Informed Consent Statement:** Not applicable.

**Acknowledgments:** I would like to thank the anonymous reviewer of my text. She/He has done a thorough job. It has led to a final result that is much improved compared to the original one. I also want to thank Rui Parada Cascais and the MDPI proofreader for the final English version of this paper.

**Conflicts of Interest:** The author declare no conflict of interest.

## Notes

1  Cf. [2] on facing emotions.
2  On experiencing *home*, cf: [4].

3    On how different traditions can come together, cf: [5].

4    On *object emotions*: [6].

5    On the effects of boredom collectively: [7]. On boredom as an ancient mood: en têi skholêi, cf: [8].

6    We will follow Heidegger's interpretation of the emotional life or attunement (Stimmung). Thence my disclaimer to the Heideggerian: I translate Stimmung as "emotion" rather than "mood" or "attunement" because I want other readers to get to one aspect of the phenomenon at stake. To the non-Heideggerian readers, I say give it a chance. I have been learning a lot about emotional life, studying Heidegger. The phenomenon which Heidegger analyses was identified by Aristotle throughout the corpus aristotelicum as *pathos* but also as *diathesis* (disposition) and *hexis* (condition or way of being (*ekhein* + adv.)). Cf.: Arist. *Metaph*. 1022b1-3: "Diathesis legetai tou ekhontos merê taxis ê kata topon ê kata dynamin ê kat' eidos: thesin gar dei tina einai, hôsper kai tounoma dêloi hê diathesis. (Disposition means arrangement of that which has parts, either in space or in potentiality or in form. It must be a kind of position, as indeed is clear from the word, disposition. Tredennick, 1933.)". In his *Nicomachean Ethics* Aristotle says that the basis of all ethical phenomena are *pathe* (affects, emotions)."Epei oun ta en têi psykhêi ginomena tria esti, pathê, dynameis, hexeis, toutôn an ti eiê hê aretê. Legô de pathê men epithymian orgên phobon, tharsos phthonon kharan philian misos pothos zêlon eleon, holôs hois hepetai hêdonê ê lupê (There are tree kinds of phenomena generated in our mind (*psykhê*): 1. An emotion (*pathos*), 2. A potentiality (*dynamis*), 3. A disposition (*hexis*). Excellence (*aretê*) must, therefore, be one of these three things. By the emotions (*pathê*), I mean desire, anger, fear, confidence, envy, joy, friendship, hatred, longing, jealousy, pity; and generally those states of consciousness which are accompanied by pleasure or pain. Rackham, 1934)." Although the philosophical tradition usually translates the word *pathos* as "affectus" and "emotions", the analyses in the *Nicomachean Ethics* and *Rhetoric* make clear that there is a much more complex dimension to the phenomenon than the linear and superficial one. For Aristotle, it is settled that 'pathê' are forms of perception: "I define *pathê*, on the other hand, as to be such phenomena as wrath, fear, shame, desire, namely such that in general are followed up by a sense of pleasure or pain [unleashed by and] in themselves (legô de pathê men ta toiauta, thumon phobon aidô epithumian, holôs hois hepetai hôs epi to polu hê aisthêtikê hêdonê ê lupê kath' hauta.)" Arist. *EE*, 1220b14-20. The mode of detecting: something sweet (*hedu*) or bitter (*luperon*) implies intentional and emotional changes that promote going after (*dioxis*) an object or running away from (*phugê*) an object. The perceptual opening (*aisthêsis*) to a real object is insufficient to "get" what in that real object brings pleasure or pain. The emotional opening clearly exceeds the reality of a thing. A brown pyramidal object is an uninteresting but clearly objective description of my favourite chocolate, Toblerone. A high-pitched sound is only frightening when it comes from the dentist's drill, for instance. Aristotle seeks to show that for every *pathos* there must be a *dynamis* as its condition of possibility. Without a *dynamis* we would not go through an emotional situation. The affective potential is waiting for the actual encounter with an object to excite an emotional response. In respect of the same content we can see opposite reactions in people. Maybe some people experience some sort of emotion and others do not. If I can sense fear, I am only afraid in a concrete frightening situation. I can be afraid of running away from some situation, though. I can be afraid of feeling shame. In that circumstance there are levels of reaction, response and behaviour that are at play. I cannot avoid feeling fear. But I can aptly respond to it through action. The *hexis* or way of being brave is based upon my resistance to fear. A courageous response allows me to overcome the pain I feel. It all begins with the condition of possibility of being able to have an emotion, sc.: fear, but also to not react to it naturally, i.e., fleeing from fear, but enduring it and waiting for the positive outcome of my action. The comparison between *hexis* and *diathesis* widens the field of understanding of emotion. According to Arist. *Metaph*. 1022b1-3, *diathesis* changes the place, the organization of the place, the appearance of the place in which I find myself. It is not only the noise of the drill that causes me to fear: it is the consulting room in that building, in that neighbourhood of the city of Lisbon, and particularly on the day of the consultation, that appears metamorphosed by the fear I have of the pain felt in the tooth when the nerve is stricken [10,11].

7    On constructing emotional past life [2].

8    Heidegger: *GA 20/30* (112) [12].

9    *Ibid*.

10   *Ibid*.

11   For the contemporary debate, see: [2,14]. Kriterium. Specifically for debating the works of Robert Solomon and Matthew Ratcliffe, cf.: [5]. The paralell drawn from the philological point of view shows clearly how different the output of different traditions can be almost juxtaposed. But the methodological approach in phenomenology is almost never taken into account. [15,16]. Most interesting for our purposes is: [17]. Capobianco presents Jung's take on the "unconscious" as "an intelligent, transpersonal structure", which "allows opposites to "happen" together and, thus, is irreducible to consciousness". Consciousness (ego) and unconsciousness "are not reducible one to the other" but "are nevertheless mutually dependent". "The unconscious maintains a primacy over consciousness, . . . can "fascinate" and "overpower" the ego; even as the unconscious maintains primacy over "subconscious" and "consciousness" Capobianco (1993) 50. Capobianco argues that Heidegger's take on ego must be understandble from his redefinition of subjectivity as Dasein, meaning that there are multiple ways in which the "I" "is". But it never gets to the bottom line of Heidegger's subjectivity as anonymous and deep. In GA vol. 20, Heidegger says that Descartes "discovered" the "ego" but has forgotten to say anything about the "sum". Implicitly in the "sum" there is a "moribundus": I am means, I'm about to die as long as I live. "The appropriate statement pertaining to Dasein in its being would have to be sum moribundus ["I am in dying"], moribundus not as someone gravely ill or wounded, but insofar as I am, I am moribundus. The MORIBUNDUS first gives the SUM its sense [Sinn]. (GA 20: 437f/317) [Translated by Theodore Kisiel]. We shouldn't forget that

"the task of the philosophers (der Philosophen Geschäft)" presented in *Being and Time* is the same in Kant's anthropology: i.e.: explicitly to "discover" "the hidden judgments of common reason (die geheimen Urteile der gemeinen Vernunft)." Heidegger (1927) [12,13].

12    Heidegger: *GA29/30* (113) [12].
13    Heidegger: *GA29/30* (114) [12].
14    *Ibid*.
15    Heidegger: *GA29/30* (115) [12].
16    *Ibid*.
17    Heidegger: *GA29/30* (117) [12].
18    *Ibid*.
19    *Ibid*.
20    Heidegger: *GA29/30* (118) [12].
21    Boredom and anxiety have the same common bottomless ground. Cf.: [18].
22    *Ibid*.
23    *Ibid*.
24    *Ibid*.
25    On depth, cf.: [19].
26    *Ibid*.
27    Heidegger. *GA29/30* (119) [12].
28    *Ibid*.
29    *Ibid*.
30    *Ibid*.
31    *Ibid*.
32    Heidegger, *GA29/30*: 120 [12].
33    *Ibid*.
34    Heidegger, GA29/30: 123 [12].
35    Heidegger stresses different aspects in this situation. He is trying to underline that emotional phenomena are experienced in the "factical life" (Faktizität). Therefore we need to avoid the unprepared interpretation that we are dealing with phenomena inside our minds (*psykhê*). He starts from the quality of "boringness" (Langweiligkeit). Cf.: GA29/30: 123: "So gehen wir schon zu Beginn absichtlich nicht von der Langeweile aus, schon deshalb nicht, weil es dann allzusehr danach aussieht, als wollten wir ein seelisches Erlebnis in unserem Bewußtsein der Analyze unterwerfen." Our underlining. We start with the boringness (Langweiligkeit): "Formal gesprochen ist die Langweiligkeit das, was etwas Langweiliges zu dem macht, was es ist, wenn es langweilend ist. (Formally formulated, the ontological quality of boredom is what makes something boring when it is boring.)." It is the experience of the active element of boredom, that which is boring, that makes any content boring [12].
36    Heidegger, *GA29/30*: 124 [12].
37    *Ibid*.
38    *Ibid*.
39    *Ibid*.
40    *Ibid*.
41    Heidegger, *GA29/30*: 126 [12].
42    Heidegger, *GA29/30*: 124: "Was das alltägliche Sprechen und Verhalten und Urteilen zum Ausdruck bringt" [12].
43    Heidegger, *GA/2930*: 125 [12].
44    *Ibid*.
45    Heidegger, *GA29/30*: 125: "Ist das so ein entsprechender Vorgang, wie wenn eintretende Kälte das Sinken der Quecksilbersäule im Thermometer verursacht? Ursache—Wirkung! Herrlich! Ist das etwa ein Vorgang, wie wenn eine Billardkugel an die andere stößt und dadurch die Bewegung der zweiten verursacht?" [12].
46    *Ibid*.
47    Heidegger, *GA29/30*: 126 [12].
48    Heidegger, *GA29/30*: 131 [12].
49    Heidegger, *GA29/30*: 127 [12].
50    Cf.: [20].
51    But there is an aspect that should be mentioned, even if it cannot be developed here. The "matter"—and indeed the "form" of which emotions are made—is musical. Music as an acoustic expression captured by acoustic perception aims at a "tonic" event that we access directly as humans, even though we may not recognise it. It is not even necessary to be a musician to have this

perception of reality. Music depends on time in order to be, just like any acoustic object. But music implies sound volume, sound quality, cadence, speed, rhythm. Life is temporal and acoustic.

52   Heidegger, *GA29/30:* 130 [12].

53   Heidgger, *GA29/30:* 131 [12].

54   Heidegger, *GA29/30:* 130 [12].

55   *Ibid*.

56   Heidegger, *GA29/30:* 122 [12].

57   *Ibid.*

58   *Ibid*.

59   Heidegger, *GA28/30:* 123 [12].

60   "The mobility and malleability of emotion are stressed by affect theorist Sara Ahmed. In the Cultural Politics of Emotion (Ahmed 2004a; 2004b), Ahmed argues that emotioin is not a thing that originates or inheres within a subject, although we often speak of it that way. Nor does it inhere within an object, waiting to be released upon contact. Instead, Ahmed understands emotion as a set of relations between subject and object that defines both. Inherently fluid and shaped by power, emotions are not psychological states for her but instead social practices. Thus, she proposes, the questioin we should ask is not what affect is but what it does: how does it circulate within a society through its circulatioijn? What sort of relations shape it or are shaped by it?" [22,23]. The analysis is obviously about a 5th century BC author. Emotions in antiquity were not just outside the subject, they were transcendence, connection to others when others appeared not as stray entities but were interpreted in the light of what they represent for us. The one we love lived for a time without existing for us as a possibility that did not even have an identity. In being loved a woman emerges as a goddess, as Aphrodite, an enemy emerges not only as a man but as the devil. Our fears and our loves do not arise only from our head but are collective entities like the boogey-man. Only modernity has confined emotions to the flow of the cogito and finds it extremely difficult to break the bonds of neuroscience.

61   Is it not emotionally that we get estranged from the world? Cf.: [24].

62   On the relation time-emotional life, cf: [25,26].

63   Cf.: [29].

64   "One feels it is boring, when on a Sunday afternoon one goes for a walk through the streets of a big city. (es ist einem langweilig, wenn man an einem Sonntagnachmittag durch die Straßen einer Großstadt geht.)" (Heidegger GA 29/30, 204) [12].

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
