# Peer review of "Paradoxes of Emotional Life: Second-Order Emotions"

_philosophies, doi:10.3390/philosophies7050109_

Round 1

Reviewer 1 Report

Interesting article to review.

However, the abstract in the introduction (above) does not match the abstract in article to be reviewed?

Can we explain an emotion applying the category of causality? Is an emotion an inner phenomenon inside our minds or exists in the outside world? Is the metaphoric use of language the only way to express emotions and feelings? How can we get to the facts of life without meaning? These questions raised are preparatory to understand the depth dimension or second-order emotions. Emotions can be anonymously present and at the same time absent? Can an emotion have its ground in the future? Is there anything outside our minds? Heidegger's phenomenological analyses of boredom (first, second orders and depth) allows putting into perspective old and new questions about emotional a priori.

YOUR ARTICLE: Heidegger explains our emotional life using three schemes: 1. causal explanation, 2. men- tal internalisation of emotions and 3. metaphorical expression. None of the three schemes explains emotion, though. Either because the causal nexus does not always occur or because objects and people in the external world are carriers of emotional agents or because language is already on a metaphorical level. But more. How is it possible that in the present there are emotions constituting our life without our being aware of their existence? From the analysis of boredom in its tree features 10 (“bored by X”, “get one self bored”, and “It is boring”) we will get at the depth dimension where 11 emotions lie, trying to wake them up and not let them go to sleep. 12 

A few more keywords would be helpful, eg. boredom, certainly. 

Author Response

Dear Reviewer, thank you so much for your helpful suggestion. 

Reviewer 2 Report

I added my comments as a separate word file. Please let me know if you do not receive the file or cannot open it. 

Author Response

Dear Reviewer, thank you so much for your helpful and insightful suggestions. I hope I've met the expectations raised by your thorough reading.

Round 2

Reviewer 2 Report

The added methodological reflections on the end of p. 3 and on p. 4 are very illuminating, as is the added discussion of contemporary literature on emotions on pp. 7 and 8, of metaphor on p. 19, and the added section 9 on emotional depth. All my main comments and suggestions were appropriately addressed.

Nevertheless, the text could still gain from minor corrections (apart from the necessary revision by someone at a native speaker level):

- It is unclear why there are now numbers throughout the abstract (maybe a typo?).

- The remarks on Aristotle on pp. 2 and 3 would perhaps fit better in the footnote 9, after mentioning Aristotle. As it is, it is somewhat puzzling the sudden discussion of pathos and Aristotle in the body of the text. Moreover, the added text repeats two of the citations in footnote 9. It would be better to rewrite the footnote to avoid repetitions.

- As footnote 14 includes many references and only some of them are discussed in more detail, it would be helpful to include “see” or “cp.” when the author is simply mentioning texts that clarify his or her views.

- The paragraph on the musical form of emotions on p. 19 (lines 758-762) should perhaps be converted into a footnote, as it reads as a brief excursus and breaks the flow of the argument.

Author Response

 Author's Notes to Reviewer in Word doc. submitted.